# Expression of Enzymes Associated with Prostaglandin Synthesis in Equine Conceptuses

**DOI:** 10.3390/ani11041180

**Published:** 2021-04-20

**Authors:** Sven Budik, Ingrid Walter, Marie-Christine Leitner, Reinhard Ertl, Christine Aurich

**Affiliations:** 1Platform for Artificial Insemination and Embryo Transfer, Department for Small Animals and Horses, Vetmeduni Vienna, Veterinärplatz 1, 1210 Vienna, Austria; marie.leitner@live.at (M.-C.L.); christine.aurich@vetmeduni.ac.at (C.A.); 2Department of Pathobiology, Institute of Anatomy, Histology and Embryology, Vetmeduni Vienna, Veterinärplatz 1, 1210 Vienna, Austria; Ingrid.walter@vetmeduni.ac.at; 3VetCore Facility for Research, Vetmeduni Vienna, Veterinärplatz 1, 1210 Vienna, Austria; Reinhard.ertl@vetmeduni.ac.at

**Keywords:** equine embryo mobility, prostaglandin synthesis, maternal recognition of pregnancy

## Abstract

**Simple Summary:**

The mobile preimplantative phase of equine gestation, taking place between day 9 and 16 after ovulation, is characterized by peristaltic contractions of the uterus caused by secretion of prostaglandins by the spheric equine conceptus. This mobility is necessary for maternal recognition of pregnancy in equids, taking place around day 14 after ovulation. The presented study investigated the spatial and temporal abundance of prostaglandin synthesis enzymes of the equine conceptus, elucidating a basal and an inducible system for prostaglandin E2. Prostaglandin F2α synthesis is restricted to the “periembryonic”pole area and relies on enzymatic conversion of prostaglandin E2. This scenario led to a model able to explain the embryonic forward motion driven by the peristaltic contractions of the uterus. In vitro incubation of primary trophoblast cell cultures with oxytocin showed no influence of this hormone on prostaglandin synthesis.

**Abstract:**

In the horse, mobility of the conceptus is required for maternal recognition of pregnancy depending on secretion of prostaglandins by the conceptus. The aim of this study was to determine the expression and localization of key enzymes of the different pathways leading to synthesis of prostaglandin E2 and F2α in the equine conceptus during the mobility phase. Enzyme expression was analyzed via quantitative RT-PCR in total RNA samples of equine conceptuses collected on days 10 (n = 5), 12 (n = 12), 14 (n = 5) and 16 (n = 7) from healthy mares. Relative abundance of cyclooxygenase (*COX*)-2 mRNA was higher (*p* < 0.05) than of *COX*-1 irrespective of conceptus age and for phospholipase A2 on day 16 in comparison to all other days (*p* < 0.01). Abundance of mRNA of cytosolic and microsomal prostaglandin E synthase (*PGES*) and of carbonyl reductase (*CBR*) 1 was not influenced by conceptus age. Immunohistochemically, COX-1, COX-2, as well as cytosolic and microsomal PGES were present in both the ectodermal and endodermal layer of the yolk sac wall. CBR-1 was restricted to periembryonic disc area. The localisation of the key enzymes explains the mechanism of embryo mobility. In vitro incubation of primary trophoblast cell cultures with oxytocin had no effect on key enzyme synthesis.

## 1. Introduction

After its descent into the uterus around day 5.5 as morula or early blastocyst, the equine conceptus undergoes a fast expansion while maintaining its spherical shape and replacing the zona pellucida by an acellular capsule (for review, see [1]). This allows for perfect adaptation of the conceptus to a phase of motility, which is extended in equine species in comparison to other domestic animal species [2]. Conceptus mobility is crucial for maternal recognition of pregnancy in horses (for review, see [1]). It depends on secretion of prostaglandin (PG) E_2_ or F_2α_ by the conceptus, which triggers peristaltic movements of the myometrium [3,4]. The production of both PGE_2_ and PGF_2α_ by equine conceptuses has been proven before [2,5], but to the best of our knowledge, a detailed analysis of the expression and localization of the enzymes involved in prostaglandin synthesis in equine conceptuses has not been performed so far.

Prostaglandin synthesis starts with the release of arachidonic acid from cell membrane phospholipids catalyzed by phospholipase A_2_ (PLA_2_) enzymes. Only cPLA_2α_ seems to be specialized in the release of arachidonic acid for eicosanoid production [6]. COX-1 and COX-2 are the enzymes catalyzing the rate-limiting step in prostaglandin synthesis, converting arachidonic acid into prostaglandin H_2_ (PGH_2_), which is then further metabolized to prostaglandin E_2_ (PGE2), PGF2α, PGD_2_ and other eicosanoids. The binding constants (*K_m_*) of the two enzymes for arachidonic acid are quite similar [6]. Due to differences in the promotor region, *COX-1* is constitutively expressed in the majority of cells, whereas *COX-2* expression is induced by multiple cytokines and growth factors. COX-2, thus, seems to be the major enzyme controlling PGE2 synthesis in response to inflammation [7]. Low, endogenous concentrations of arachidonic acid are metabolized by COX-2, whereas higher concentrations of exogenous origin are converted by COX-1 [8]. Immunofluorescence studies revealed that both isoenzymes are localized in the endoplasmic reticulum and the nuclear envelope [9,10].

Lim and co-workers [11] demonstrated by targeted disruption of the *COX-2* gene in mice that absence of *COX-2* expression causes multiple female reproductive failures affecting ovulation, fertilisation, implantation and decidualisation. None of these effects were observed in mice with a disrupted *COX-1* gene.

PGH_2_ produced by cyclooxygenase isoenzyme serves as substrate for prostaglandin E2 and F2_α_ synthases (PGES, PGFS). Since they catalyse the generation of the final active product, they are termed terminal synthases. Studies elucidated at least three distinct prostaglandin E synthases (PGES) and functional coupling between particular COX and PGE*S* isoforms, which are: (a) microsomal prostaglandin E synthase-1 (mPGES-1), (b) cytosolic prostaglandin E synthase (cPGES) and (c) microsomal prostaglandin E synthase-2 (mPGES-2). Induction of PGE2 synthesis seems to depend on concordantly increased *mPGES-1* and *COX-2* gene expression, which refers to functional coupling of the two enzymes [12]. PGF2α is synthesized mainly by means of the terminal synthase PGFS, using PGH2 as substrate [13], but there are also alternative synthesis pathways mediated by the enzymes aldo-keto reductase family 1 member C1 and C2 (AKR1C1 and AKR1C2) as well as carbonyl reductase 1 (CBR-1), using PGE2 as substrate [14].

The aim of the present study was to investigate the expression and localization of key enzymes of the different pathways leading to the synthesis of PGE2 and PGF2_α_ in order to detect the main synthesis pathway used by the early equine conceptus during the mobility phase, i.e., between days 10 and 16 after ovulation. The effect of oxytocin on trophoblast PGE2 synthesizing enzymes was also investigated in primary cell cultures derived from the yolk sac wall of equine embryos at days 10, 12, 14 and 16. Detailed knowledge of the equine conceptus’ prostaglandin synthesis will help understand the physiology of maternal recognition of pregnancy in this species.

## 2. Materials and Methods

### 2.1. Animals and Conceptus Collection

The investigation was approved by the institutional ethics and animal welfare committee in accordance with GSP guidelines and national legislation. Conceptus production and collection was done according to the Austrian law for animal experiments in studies performed previously (permit numbers BMBWK-68.205/0023-BrGT/2007 and GZ 68.205/0078-WF/II/3b/2014). Eight healthy fertile mares (6 Haflinger and 2 Lipizzaner, aged 4 to 16 years) were used. Oestrous mares were inseminated with raw semen from stallions of proven fertility at 48 h-intervals until detection of ovulation by transrectal ultrasound examination, which was done at 24 h-intervals. The day of detection of ovulation (disappearance of the preovulatory follicle) was defined as day 0. Conceptus collection was performed as described previously [15].

Conceptuses were washed in pre-warmed PBS for removal of any attaching cells and scored for quality and size. For RT-PCR, part of the yolk sac wall of the conceptus was excised with a scalpel blade, transferred with a small amount of PBS into an Eppendorf tube, shock frozen in liquid nitrogen and stored at −80 °C until RNA preparation. For immunohistochemistry, the remaining conceptuses were placed into 4% neutral buffered formaldehyde for fixation.

### 2.2. Extraction of RNA

After homogenisation of frozen single conceptuses or tissue samples with a sterile glass homogenizer, total RNA was extracted using TriReagent (Sigma-Aldrich, Vienna, Austria) according to the manufacturer’s protocol. The total RNA sample was DNase I RNase free (Thermo Fisher Scientific, Waltham, MA, USA) treated in order to remove putative traces of genomic DNA. Quantification and the 260/280 nm ratio of total RNA was performed by use of a Tray cell (Hellma, Jena, Germany) and Bio Photometer (Eppendorf Austria, Vienna, Austria). Only samples showing expected 260/280 nm ratio values near 2 were used for qualitative RT-PCR.

### 2.3. Reverse Transcription Qualitative PCR (RT-PCR)

RT-PCR primers: Primer sequences for qualitative PCR were designed from sequences obtained from PubMed nucleotide search (https://www.ncbi.nlm.nih.gov/nucleotide/ accessed on 23 October 2012) by use of the GCG program “prime” of the Wisconsin package [16]. Preferable primer pairs resulting in intron spanning products binding in conserved regions of the sequences were selected in order to optimize fitting for equine sequences and allowing distinction between cDNA derived and genomic products (Table 1).

RT-PCR: 50 ng of total conceptus RNA was reversely transcribed using specific primers designed from equine sequences (Table 1). The reaction was performed using a one-step RT-PCR kit (Qiagen, Venlo, The Netherlands) according to the description of the manufacturer, using 30 pmol of each primer and more than 1 pg of total RNA. The cycling profile was 30 min at 50 °C (reverse transcription), 10 min at 95 °C (polymerase activation), 35 cycles 94 °C for 1 min, 52 °C for 1 min, 72 °C for 1 min for PCR amplification; 72 °C 10 min for final extension and final storage at 4 °C overnight.

### 2.4. Sequencing of RT-PCR Products

Products amplified by specific primers were separated by electrophoresis on 1.5% agarose gel containing ethidium bromide using TAE (Tris/acetate/ETDA) buffer (1×: 40 mM Tris (pH 7.6), 20 mM acetic acid, 1 mM *EDTA* and 6× loading dye (Fermentas, Vilnius, Lithuania). After UV detection by means of ChemiDoc (Biorad Laboratories GmbH, Vienna, Austria), bands of interest showing the expected length were excised, cleaned with a Qiaex II kit (Qiagen) and sequenced by commercial sequencing service using the primers for amplification (IBL, Vienna, Austria). Sequences were compared with sequences from the database using standard nucleotide blast search (http://www.ncbi.nlm.nih.gov/blast/, accessed on 5 January 2014).

### 2.5. Reverse Transcription Quantitative PCR (qRT-PCR)

Total RNA samples were treated with DNase I using the Turbo DNA-free kit (Thermo Fisher, Waltham, MA, USA) following the recommended protocol. RNA concentrations were measured on a NanoDrop 2000 c UV spectrophotometer (Thermo Fisher), and RNA integrity numbers (RINs) were determined on an Agilent 2100 Bioanalyzer using an RNA 6000 Nano Kit (Agilent Technologies, Santa Clara, CA, USA). The RIN values ranged from 8.8 to 10. For qRT-PCR, 500 ng of total RNA was reverse transcribed into cDNA with a High Capacity cDNA Reverse Transcription Kit (Thermo Fisher). qRT-PCR assays were designed using the PrimerQuest design tool (Integrated DNA Technologies; http://eu.idtdna.com/PrimerQuest/Home/Index, accessed on 13 February 2015) or taken from the literature [17]. Assay details are listed in Table 2.

The qRT-PCR was done in 20 µL reactions including 20 ng cDNA, 0.2 mM of each dNTP, 3 mM MgCl_2_, 1× buffer B2 (Solis BioDyne, Tartu, Estonia), 300 nM of each primer, 200 nM probe, 50 nM ROX reference dye (Biotium, Hayward, CA, USA) and 1 unit of HOT FIREPol DNA polymerase (Solis BioDyne). All samples were analyzed in triplicates on a Viia7 Real-Time PCR System (Life Technologies, Carlsbad, CA, USA). Thermal cycling conditions were as follows: initial denaturation at 95 °C for 10 min, followed by 45 cycles of 95 °C for 15 s and 60 °C for 1 min. Fluorescence signals were collected during the 60 °C amplification step. Two candidate reference genes (RGs), *ACTB* and *RPL4*, were included for normalization. The expression stability of the two RGs was assessed using the RefFinder analysis tool [18], identifying *ACTB* as the more stably expressed gene. Target gene expression levels were normalized to those of ACTB, and relative expression changes were calculated using the comparative 2^-ddCT method [19].

### 2.6. Immunohistochemistry

Equine conceptuses from days 10, 12 and 14 (n = 2 from each) previously embedded in Histo Comp (Vogel, Giessen, Germany) were cut at 3 µm. Equine conceptuses at day 16 could not be flushed in quality suitable for immunohistochemistry. Endogenous peroxidase activity was blocked by incubation of the sections in 0.6% H_2_O_2_ for 15 min. Incubation in 1.5% normal goat or rabbit serum, respectively, was used to minimize nonspecific antibody binding. Subsequently, slides were incubated with the primary antibody overnight at 4 °C. Polyclonal antibodies directed against COX-1, COX-2 (Santa Cruz Biotechnology, Santa Cruz, CA, USA), COX-1 (C20): sc-1752, dilution 1:500; COX-2 (C-20): sc-1745, dilution 1:500, both from goat), previously proven to work in the horse [20], and against mPGES-1 and cPGES, proven to be specific to equine [21] (Cayman Chemical Company, Ann Abor, MI, USA), Prostaglandin E Synthase-1 (microsomal) polyclonal antibody # 160140, diltution 1:1:150; Prostaglandin E Synthase (cytosolic) polyclonal antibody # 160150, dilution 1:1000, both rabbit), as well as CBR-1 (Thermo Fisher Scientific, # PA5-53799, polyclonal, rabbit, dilution 1:300) were used as primary antibodies. Heat antigen retrieval was done in EDTA buffer pH 9 for COX-1 and COX-2 and citrate buffer pH 6 for CBR-1. A biotinylated anti-goat antibody (Vector Laboratories Inc., Burlingame, CA, USA) was used as secondary antibody for COX-1 and COX-2, a Vectastain ABC kit (Vector Laboratories) was utilized to enhance binding signals and for horseradish peroxidase coupling. For PGES and CBR-1 detection, a poly-HRP (horseradish peroxidase) anti-rabbit IgG (Powervision, ImmunoVision Technologies, Daly City, CA, USA) was used as secondary antibody. Subsequently, sections were washed and developed in DAB substrate (′3′3 Diaminobenzidine, Sigma; 10 mg/50 mL in 0.1 M Tris buffer, pH 7.4 and 0.03% H_2_0_2_) for 10 min at room temperature. After washing, the sections were counterstained with hematoxylin, dehydrated and mounted in DPX (Fluka, Buchs, Switzerland). Negative controls were incubated with buffer instead of primary antibody. Positive controls included horse tissue sections of suitable organs.

### 2.7. Primary Trophoblast Cell Culture

Bilaminar trophoblast pieces from day 12, 14 and 16 equine conceptuses were selected for establishment of primary cell cultures (n = 6). Day 10 equine conceptuses might be unilaminar depending on the size and were therefore not used. After washing several times in sterile pre-warmed Ca/Mg-free PBS (#70011 Gibco, Thermo Fisher Scientific), the pieces were transferred into 4-well plates (Nunc, Thermo Fisher Scientific), adding 1 mL cell culture medium (D-MEM/High Glucose (500 mL, +4.00 mM l-glutamine, +4500 mg/L glucose, +sodium pyruvate) (GE Healthcare Life Science, Marlborough, MA, USA), 10% horse serum (B11-124, PAA Laboratories, Pasching, Austria), 0.05 % penicillin-streptomycin (PreMix, Roth, Germany), 0.05% nonessential aminoacids (100 mL, Gibco, Thermo Fisher Scientific), 0.05 % insulin-transferrin-selenium (10 mL, Gibco, Thermo Fisher Scientific). Soft tissue was dispersed in the culture medium by pipetting up and down using a 1000 µL pipette. It was subsequently incubated (Galaxy S+ RS Biotech, Irvine, UK) at 38 °C with 5% CO_2_ in air. After 48 h of culture, 4-well plates were checked for attachment and growth of the cells. After reaching confluence, the trophoblast cells were transferred to Nunclon Delta flasks (Nunc EasYFlask 25 cm^2^, Thermo Fisher Scientific) with 3 mL cell culture medium. Cells were detached with a cell scraper (REF 353085, BD Falcon, Franklin Lakes, NJ, United States) and transferred into a 15 mL tube (cell star tube, Greiner, Kremsmünster, Austria) and centrifuged for 10 min at 150 *g*. After centrifugation, the supernatant was discarded, and cells were resuspended with fresh medium and again incubated at 38 °C with 5% CO_2_ in air. When the trophoblast cells reached confluence again, cells were detached, and the suspension was split into two equal parts. Cells were centrifuged at 150 *g* and the supernatant was discarded. Each suspension was transferred into 4-well plates. After growing, attaching and reaching confluence, the cells were washed twice with PBS without Ca/Mg, and the cell culture medium was changed to inactivated medium heated previously to 65 °C for 30 min in order to inactivate any endogenous oxytocin [22]. Whereas half of the wells were incubated with medium containing 0.318 nM oxytocin (03251-500IU, Sigma, St. Louis, MO, USA; group oxytocin), the other half was incubated with medium without oxytocin and served as control. All tissue cultures were then incubated for 24 h at 38 °C with 5% CO_2_ in air. Subsequently, cells were mechanically detached and centrifuged in an Eppendorf tube for 10 min at 300× *g*. The supernatant was discarded, the pellet shock-frozen in liquid nitrogen and stored at −80 °C until RNA extraction.

### 2.8. Statistical Analysis

Statistical analysis was performed with IBM SPSS statistics software (version 24.0; Armonk, NY, USA). Data were tested for normal distribution by Kolmogorov–Smirnov test. Results from quantitative PCR were compared by nonparametric Kruskal–Wallis test followed by Mann–Whitney test in case of significant differences. For all statistical comparisons, a *p*-value < 0.05 was considered significant.

## 3. Results

### 3.1. Qualitative RT-PCR for Enzymes Involved in PG Synthesis in Equine Conceptuses

RT-PCR was performed from total RNA of conceptuses collected on day 10, day 12 and day 14 (n = 1 per day): In all conceptuses, specific amplicons for *PLA2*, *cPGES*, *COX-2* and *mPGES-1* were detected. Very weak amplification for *PGFS* were present in the day 10 and 14 conceptuses, but not in the day 12 conceptus. Weak amplicons for *COX-1* were detected in all conceptuses irrespective of collection day (Figure 1).

### 3.2. Quantitative RT-PCR (qRT-PCR) Analysis for Enzymes Involved in PG Synthesis in Equine Conceptuses

*PLA2*, *COX-1*, *cPGES*, *COX-2*, and *mPGES-1* amplifications were detected in total RNA samples of all equine conceptuses collected on day 10 (n = 5), 12 (n = 12), 14 (n = 5) and 16 (n = 7). Higher abundance of *PLA2* was observed on day 16 than on days 10, 12 or 14 (*p* < 0.01; see Figure 2). The *cPGES* and *mPGES-1* transcripts were consistently expressed in all samples. *PGFS* was either not detectable or below the quantification limit of qRT-PCR and therefore excluded from quantitative analysis. The abundance of *COX-2* was higher (*p* < 0.05) than that of *COX-1* (20- to 75-fold, at day 10 to day 16) irrespective of conceptus age. The relative *COX-2* abundance continuously increased and was higher (*p* < 0.05) in day 12, 14 and 16 conceptuses than in day 10 conceptuses (see Figure 2).

In a second experiment using the same RNA preparations, the abundance of *AKR1C1* and *CBR-1* transcripts was analyzed in order to verify possible synthesis *of PGF2α* from *PGE2*. *AKR1C1* could not be detected in any conceptus sample, whereas CBR-1 was present in all samples independent of collection day (no differences between days; see Figure 2).

### 3.3. Quantitative RT-PCR Analysis of Enzymes Involved in Prostaglandin Synthesis in Primary Equine Trophoblast Cell Cultures With and Without Oxytocin

Although transcripts of all prostaglandin E2 synthesis enzymes were detected, the presence of oxytocin in the culture medium did not alter the relative amplification of any enzyme examined (see Figure 3).

### 3.4. Immunohistochemistry

The COX-1 protein was detectable in all conceptuses examined by immunohistochemistry irrespective of age (n = 8). The abundance was similar among the different conceptus tissues (trophectoderm, endoderm (Figure 4A–C), hypoblast, epiblast and Rauber’s layer (Figure 5A,B). Staining within cells appeared homogenous.

Immunoreactive COX-2 was more abundant in the trophectoderm than in the endoderm of conceptuses collected on days 10, 12 and 14 (Figure 6A–C). On day 10, staining of the epiblast and hypoblast was weaker compared to that of Rauber´s layer (Figure 5C). On day 12, only a thin coloured line was still visible on the top of the epiblast, which might have consisted of remnants from Rauber´s layer (Figure 5D). Within cells of the trophectodermal layer, COX-2 immunoreactive protein exhibited perinuclear and apical hotspots.

On days 10 and 12, cPGES protein was equally distributed in the trophectodermal and endodermal cell layer. On day 14, staining of the endodermal layer appeared weaker than that of the trophectodermal layer (Figure 4D–F). The mPGES-1 staining was equally distributed between trophectoderm and endoderm of the yolk sac wall (Figure 6E–G). On day 10, CBR-1 protein was scarcely detectable in the trophectoderm by immunohistochemistry, but much more pronounced in the endoderm (Figure 7A). On day 12, the trophectoderm was unstained, but epi- and hypoblast showed intense staining (Figure 7B,C). Single scattered trophectodermal cells in the epiblast showed no staining (arrow, Figure 7C). The endoderm of the yolk sac wall appeared differential in abundance of immunoreactive protein: close to the hypoblast, staining was most intense. In the endodermal layer, staining lost intensity with increasing distance from the hypoblast (Figure 7B,C). On day 14, *CBR-1* protein was almost absent in the trophectoderm, whereas the yolk sac endoderm and the epiblast were considerably stained (Figure 7D,E).

## 4. Discussion

In this investigation, a detailed analysis into prostaglandin synthesis in equine conceptuses collected around the time of maternal recognition of pregnancy was performed for the first time. Our results demonstrate that equine conceptuses possess an inducible as well as a basal PGE_2_ synthesis system. The inducible system is represented by cytosolic cPLA2, COX-2 and mPGES-1 localized in the yolk sac wall, namely, trophectoderm and yolk sac endoderm. COX-2 and mPGES-1 are co-localized in these tissues, suggesting an effective transfer of the unstable PGH2 [23]. The enzyme mPGES-1 is the only PGE synthase induced by proinflammatory stimuli, including lipopolysaccarids (LPS), tumor necrosis factor α (TNFα) and IL-1β [23]. This conformity with COX-2 and the spatial and temporal co-expression make them a functionally coupled inducible PGE2 synthesis system [24]. Early growth response (EGR)-1, an inducible zinc finger protein, binds to the proximal GC box in the *mPGES-1* promoter region and facilitates the inducible transcription of the *mPGES-1* gene [25]. EGR-1 was proven to be the most abundant non-ribosomal transcript in pregnant equine endometrium at day 9 [26] and was still strongly expressed at the time of maternal recognition of pregnancy [27]. Most likely, the equine conceptus’ PGE2 synthesis can be influenced by uterine factors like EGR-1 or by itself in autocrine loops, similar to the pig [28,29]. The intensity of prostaglandin E2 production may have a pivotal role for the equine conceptus during its mobile phase of gestation: On the one hand, it mediates conceptus mobility, proven to be important for maternal recognition of pregnancy (reviewed by Aurich and Budik [1]). On the other hand, its production influences the relation between luteotrophic PGE2 [30] and luteolytic PGF2α [31], which might be important for maintenance of pregnancy in the equine species, as has been proven in the pig [28,29]. In this context, the absence of the terminal PGF2α synthase in equine embryos as well as the absence of AKR1C1 is remarkable. Our results suggest that equine conceptuses produce PGF2α from PGE2 via CBR-1, which was abundant in all conceptus stages examined. In the yolk sac endoderm of day 12 equine conceptuses surrounding the embryonic disc, expression of all enzymes involved was detected (COX-2, mPGES-1 and CBR-1), making it the most probable location for PGF2α synthesis. This suggests a polar (“periembryonic disc”) production of PGF2α in equine conceptuses, which might be crucial for its migration within the uterus (Figure 8).

As expected, *COX-1* expression was abundant in all tissues at all times investigated, confirming its role as a constantly expressed gene. The cPGES is only capable of converting COX-1-, but not COX-2-derived PGH2 into PGE2 [32]. In the equine conceptuses included in the present study, we detected co-localisation of these two enzymes (COX-1 and cPGES), suggesting a functional dependence. Incubation of primary trophoblast cell cultures with oxytocin resulted in no change of mRNA expression of any enzyme investigated. Therefore, we can conclude that oxytocin had no direct influence on transcription of enzymes associated with PGE2 and PGF2α synthesis at least in the trophoblast in vitro. However, this might be differently regulated on the protein or activity level.

## 5. Conclusions

In the present study, our results revealed two independent prostaglandin E2 producing systems in equine conceptuses, namely PLA2/COX-2/mPGES-1, which could be induced by its own PGE2 production by an autocrine loop but also by endometrial factors like EGR-1 or TNFα [27,33], and PLA2/COX-1/cPGES as a constant basal one.

The production of PGF2α on the animal pole of the conceptus induced a polarity in PGF2α and PGE2 abundance on the embryo surface, which may explain how myometrial peristaltic movements result in forward conceptus mobility.

## Figures and Tables

**Figure 1 animals-11-01180-f001:**
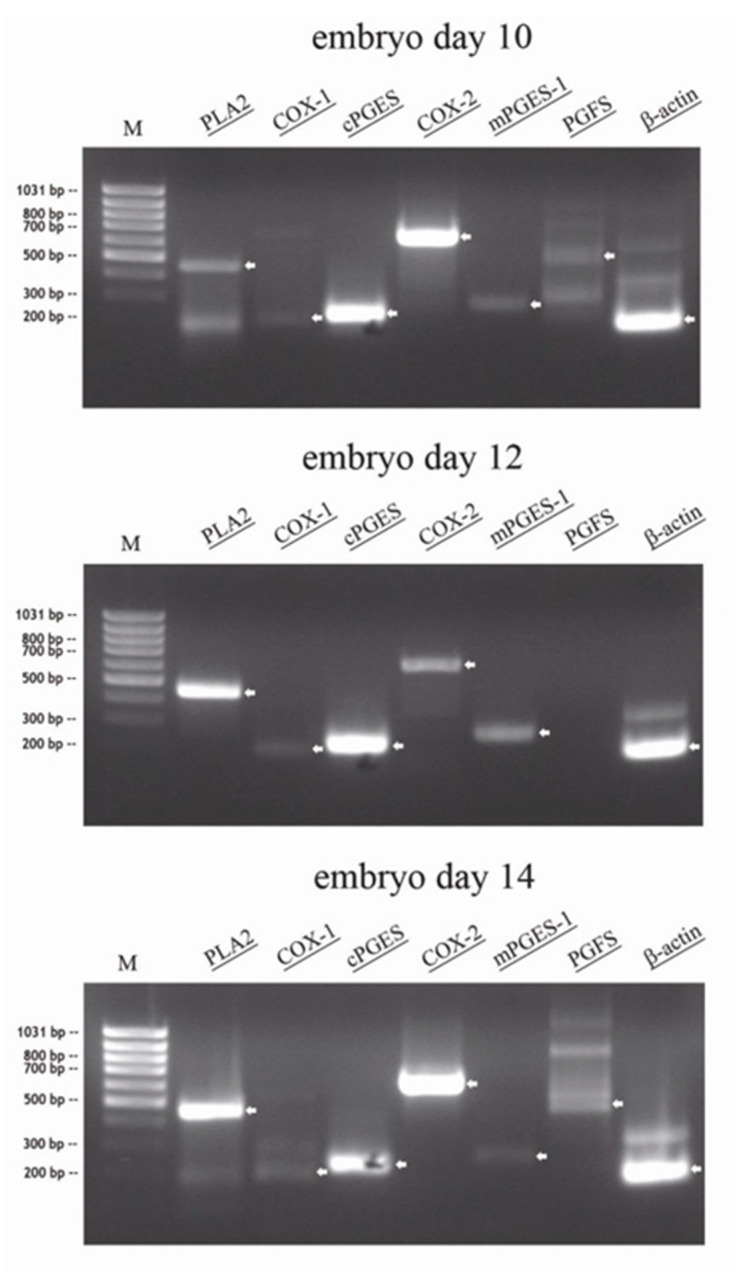
RT-PCR of total RNA of different equine embryos (n = 1) at days 10, 12 and 14, separated on a 1.5% agarose gel, visualized by UV after ethidium bromid staining (specific bands marked by arrow: *PLA2*: 463 bp, *COX-1*: 204 bp, *cPGES*: 229 bp, *COX-2*: 627 bp, *mPGES-1*: 263 bp; *PGFS*: 480 bp, *β-actin*: 190 bp).

**Figure 2 animals-11-01180-f002:**
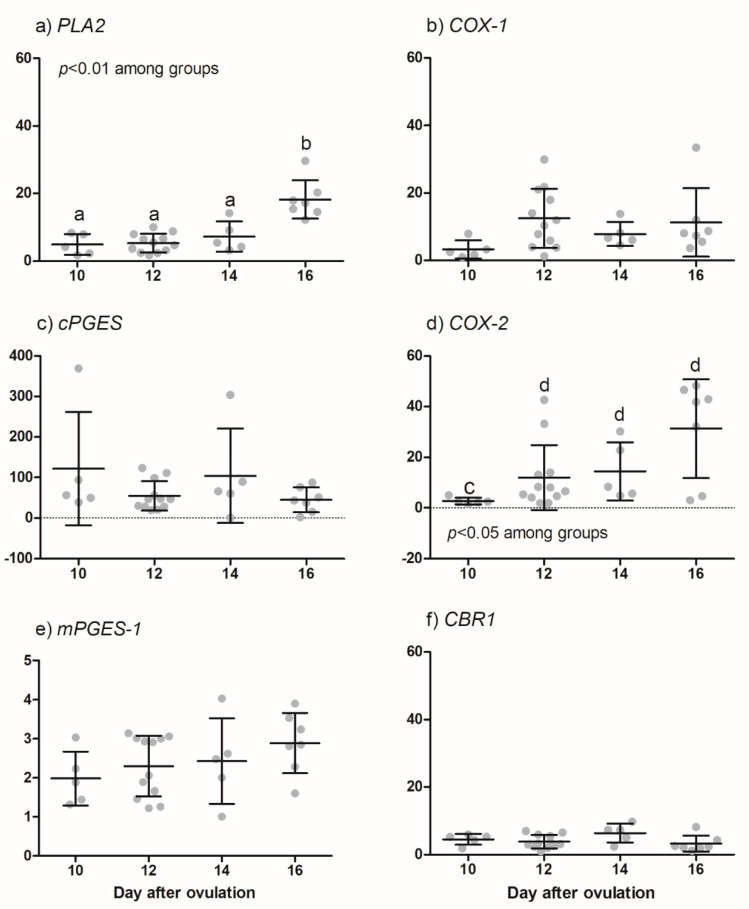
Quantitative RT-PCR (qRT-PCR) analysis for enzymes involved in prostaglandin synthesis of cDNA derived from equine conceptuses collected on day 10 (n = 5), 12 (n = 12), 14 (n = 5) and 16 (n = 7). Data are shown as individual relative amplification together with the respective mean ± SD. Different small letters correspond to significant differences (a, b: *p* < 0.01, c, d: *p* < 0.05).

**Figure 3 animals-11-01180-f003:**
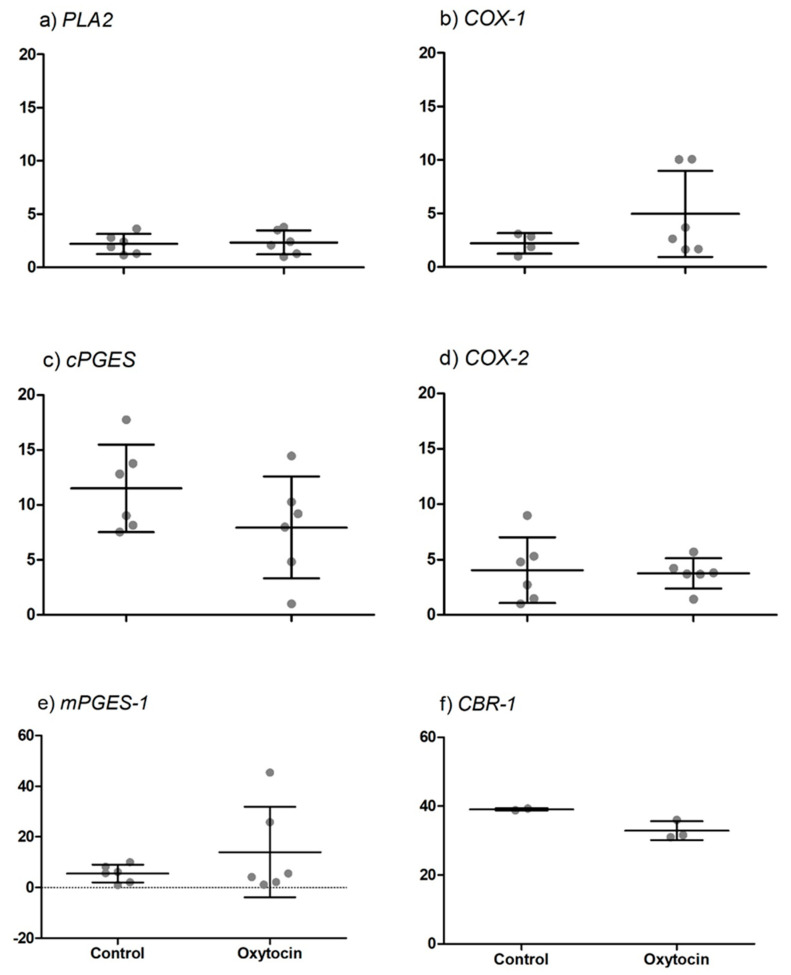
Relative quantitative RT-PCR (qRT-PCR) analysis for enzymes involved in prostaglandin synthesis determined in cDNA derived from primary trophoblast cell cultures (n = 6) incubated with or without oxytocin. Data are shown as individual relative amplification together with the respective mean ± SD. No significant differences between treatments.

**Figure 4 animals-11-01180-f004:**
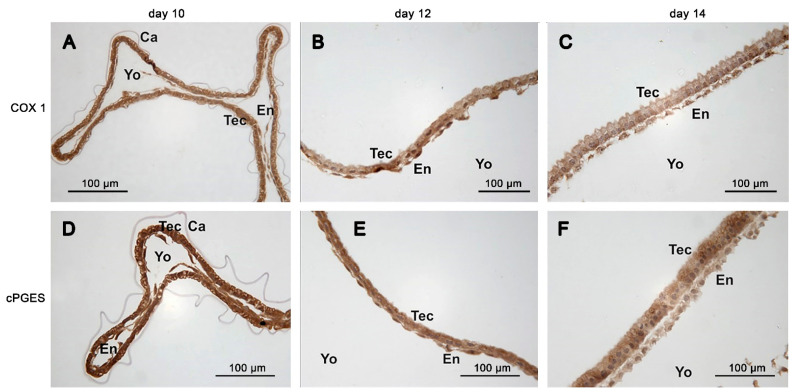
Immunohistochemistry of the basal prostaglandin E2 forming enzymes in equine embryos at days 10, 12 and 14 after ovulation (**A**,**D**) day 10; (**B**,**E**) day 12; (**C**,**F**) day 14: (**A**–**C**): COX-1; (**D**–**F**): cPGES; (Ca: capsule, En: endoderm, Tec: trophectoderm, Yo: yolk sac).

**Figure 5 animals-11-01180-f005:**
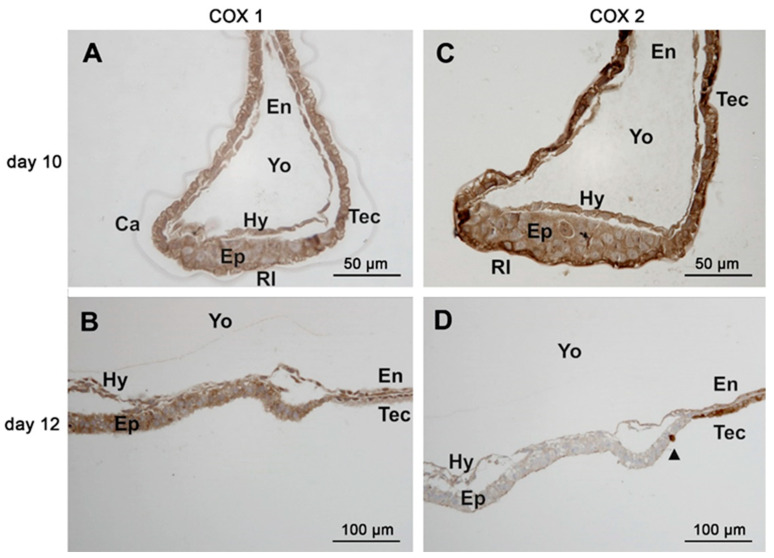
Immunohistochemistry of cyclooxygenase enzymes of the inner cell mass/embryonic disc on days 10 and 12 after ovulation (day 10: (**A**,**C**); day 12 (**B**,**D**)); (**A**,**B**): COX-1, (**C**,**D**)*:* COX-2 (Ca: capsule, En: endoderm, Ep: epiblast, Hy: hypoblast, Rl: Rauber´s layer, Tec: trophectoderm, Yo: yolk sac). The arrowhead in D is indicating a scattered trophectodermal cell within the epiblast.

**Figure 6 animals-11-01180-f006:**
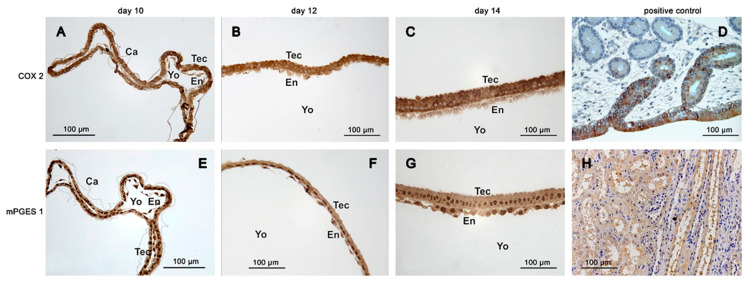
Immunohistochemistry of the inducible prostaglandin E2 forming enzymes in equine embryos at days 10, 12 and 14 after ovulation (**A**,**E**) day 10; (**B**,**F**) day 12; (**C**,**G**) day 14: (**A**–**C**): COX-2; (**D**): positive control equine endometrium COX-2; (**E**–**G**): mPGES-1; (**H**): positive control equine kidney mPGES-1; (Ca: capsule, En: endoderm, Tec: trophectoderm, Yo: yolk sac).

**Figure 7 animals-11-01180-f007:**
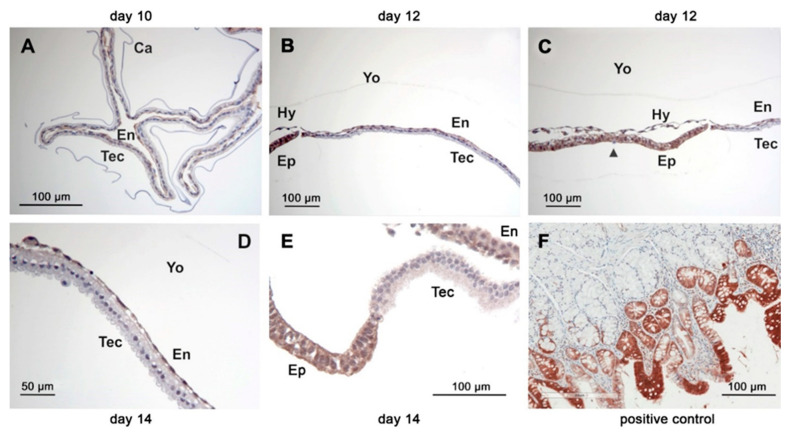
Immunohistochemistry of equine embryos at days 10 (**A**), 12 (**B**,**C**) and 14 (**D**,**E**) after ovulation using a polyclonal CBR-1 antibody. Positive control equine small intestine (**F**). (Ca: capsule, En: endoderm, Ep: epiblast, Hy: hypoblast, Tec: trophectoderm, Yo: yolk sac). The arrowhead in D indicates a scattered trophectodermal cell within the epiblast.

**Figure 8 animals-11-01180-f008:**
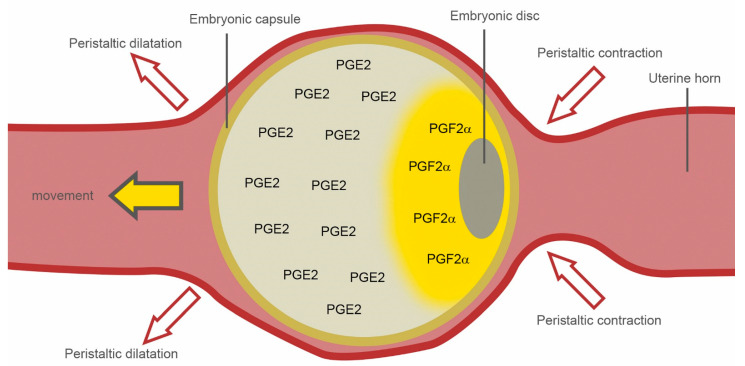
Suggested model for explaining how PGE2 and PGF2α production localized at two opposite poles of the equine conceptus facilitates equine conceptus mobility: PGE2 production at the left (“a-embryonic”) pole induces myometrial dilatation, while PGF2α production at the opposite right (“embryonic”) pole induces peristaltic contractions of the myometrium.

**Table 1 animals-11-01180-t001:** Primers used for qualitative RT-PCR.

Accession number	Gene Symbol	Gene Name	Oligo	Sequence (5′–3′)	Amplicon Length
AF092539	*PLA2*	Equus caballus Phospholipase A2	Forward Reverse	TGGGAGAGAAGAAAGAGG GTGAGAAGTAAAAGGGGG	463 bp
AB039865	*COX-1*	Equus caballus Cyclooxygenase 1	Forward Reverse	ACCCCAGAACCAGATGGC TACTCCTCAATCACGATCTTG	204 bp
AB041771	*COX-2*	Equus caballus Cyclooxygenase 2	Forward Reverse	CTGAGCACCTGCGGTTTG CAGCCATTTCCTTCTCTCC	627 bp
XM_001492303	*cPGES*	Equus caballus Cytosolic Prostaglandin E Synthase	Forward Reverse	CGAAAAGGAGAATCTGGC CGTCATCACTGTCTTGTGAATC	229 bp
AY057096	*mPGES-1*	Equus caballus Membrane bound (microsomal) Prostaglandin E Synthase-1	Forward Reverse	GCTGCTGGTCATCAAGATGT CCCAGGAAGAAGACGAGAAA	263 bp
AY304536	*PGFS*	Equus caballus Prostaglandin F Synthase	Forward Reverse	CATCCGAAGCAAAATTGAAG AGACGTTGAGTCCCCAAAG	480 bp
AF035774	*β-ACT*	Equus caballus Beta Actin	Forward Reverse	ATGGAATCCTGTGGCATC GCGCAATGATCTTGATCTTC	190 bp

**Table 2 animals-11-01180-t002:** Primer and probes used for qRT-PCR.

Accession Number	Gene Symbol	Gene Name	Oligo	Sequence (5′–3′)	Amplicon Length	PCR Efficiency (%)	Reference
NM_001081838.1	*ACTB*	Equus caballus beta actin	Forward Reverse Probe Probe	CCGGGACCTGACGGACTA CCTTGATGTCACGCACGATT FAM-TACAGCTTCACCACCACGGCCG-BHQ1	95 bp	0.96	Herrera-Luna et al., 2016 [17]
XM_023631879.1, XR_002804825.1, NM_001081908.1	*AKR1C1*	Equus caballus aldo-keto reductase family 1, member C1	Forward Reverse Probe	CCTAAACCGAAATCTGCGATATG CTAGTGGAGATCAGGACAAAGG FAM-ATGTTTGCTGGCCACCCTGAGTAT-BHQ1	109 bp	0.95	-
XM_001916273.4, XM_001493545.5	*CBR1*	Equus caballus carbonyl reductase 1	Forward Reverse Probe	GAAGTGACAATGAAAACAAACTTT CTAGACACATTCACCACTCTGC FAM-GCACGGAGCTACTGCCTCTCATAAA-BHQ1	98 bp	0.90	-
NM_001163976.1	*COX-1* *(PTGS1)*	Equus caballus prostaglandin-endoperoxide synthase 1	Forward Reverse Probe	GCGCTGGTTCTGGGAATTT TAAGGTTGGAACGCACTGTGAGT FAM-CAACGCCACCTTCATCCGTGACATG-BHQ1	83 bp	0.92	-
NM_001081775.2	*COX-2* *(PTGS2)*	Equus caballus prostaglandin-endoperoxide synthase 2	Forward Reverse Probe	GAGGTGTATCCGCCCACAGT AGCAAACCGCAGGTGCTC FAM-TCAGGTGGAAATGATCTACCCGCCTCA-BHQ1	81 bp	0.93	-
XM_001492303.4	*cPGES* *(PTGES3)*	Equus caballus prostaglandin E synthase 3 (cytosolic)	Forward Reverse Probe	TTTGCACTGTCAGTATGGCAAGT AATCCCCACTGTCATGAACATAAA FAM-TTCTTTCTTTGATTCAGCTTGTACTGCGGC-BHQ1	127 bp	0.95	-
AY057096.1, NM_001081935	*mPGES-1* *(PTGES)*	Equus caballus prostaglandin E synthase (microsomal)	Forward Reverse Probe	AGTTTCACCGGGACGACCA AGTAGACGAGGCCCAGGAACA FAM-CGTGGAGCGTTGCCTGAGAGCC-BHQ1	99 bp	0.91	-
XM_001497094.3	*RPL4*	Equus caballus ribosomal protein L4	Forward Reverse Probe	CTGTGTTCAAGGCTCCCATTC GCACTGGTTTGATGACCTGCTA FAM-AACAGACAGCCTTATGCCGTCAGCG-BHQ1	115 bp	0.97	-

## Data Availability

Data can be made available on request.

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
