# Peer review of "Expression of Enzymes Associated with Prostaglandin Synthesis in Equine Conceptuses"

_animals, 2021, doi:10.3390/ani11041180_

Round 1
Reviewer 1 Report
Dear authors,
This study analyses the gene expression and localization of different enzymes related to synthesis of prostaglandins in the equine conceptuses at days 10, 12, 14, and 16 of gestation. The manuscript is well structured and written, the introduction provides sufficient background, the research design is appropriate, the methods and the results are correctly described and presented, and the conclusions are supported by them. However, major revisions are necessary:
- “in vitro” should write in italics.
- In 2.2 section, the authors should indicate the values of integrity and quantity of RNA extracted, and the exact amount of RNA used for RT-PCR should explain.
- In Figure 2d, change the colour of 12 days (is not the same colour than the others).
- Figure 6 is missing. Please, add it.
I hope these recommendations improve the quality of manuscript.
Author Response
Dear authors,
This study analyses the gene expression and localization of different enzymes related to synthesis of prostaglandins in the equine conceptuses at days 10, 12, 14, and 16 of gestation. The manuscript is well structured and written, the introduction provides sufficient background, the research design is appropriate, the methods and the results are correctly described and presented, and the conclusions are supported by them.
Thank You for the positive comments!
However, major revisions are necessary:
- “in vitro” should write in italics. All were changed into italics!
- In 2.2 section, the authors should indicate the values of integrity and quantity of RNA extracted, and the exact amount of RNA used for RT-PCR should explain. To evaluate our total RNA samples we performed a spectrometric analysis using a try cell and a biophotometer. The amounts for qualitative RT-PCR were calculated according to these mesurements. Only RNA samples showing a 260/280 nm ratio around 2 were used for RT-PCR.
- In Figure 2d, change the colour of 12 days (is not the same colour than the others). Thank You we changed the colour!
- Figure 6 is missing. Please, add it. I really beg Your pardon but I had severe problems by sending the first manuscript due to inappropriate internet quality! That was the reason for that failure!
I hope these recommendations improve the quality of manuscript.
We hope that our ameliorations make it possible for publication in “Animals”.
Happy Eastern!

Reviewer 2 Report
The draft “Expression of enzymes associated with prostaglandin synthesis 2 in equine conceptuses” by Budik et al. addresses a very interesting and important topic in equine maternal recognition. However, important data are not presented in the results, namely the description of figure 5, and figure 6 is not in the submitted draft thought described in the text.
Additionally it would improve data presentation if on figures 4, 5 and 6 authors could consider adding
- at the top of the “columns” which day/gene they correspond to;
- at the left side of the “lines” which day/gene they correspond to.
It would also help readers if those are consistent, meaning either always days at the top or at the side.
By reading, the abstract and the Introduction readers get confused:
What was indeed used for each experiment: equine conceptuses collected on days 10 (n=5), 12 (n=12), 14 (n=5) and 16 (n=7) from healthy mares for enzyme expression, and primary cell cultures derived from the yolk sac wall of equine embryos at days 10, 12, 14 and 16 to investigate the effect of oxytocin on trophoblast PGE2 synthesizing enzymes? Or both? Please clarify.
Did the authors perform different RNA extraction for Reverse transcription qualitative PCR (RT-PCR) and Reverse transcription quantitative PCR (qRT-PCR)? If not lines 141-145 should be moved to 2.2 Extraction of RNA.
It is not explained why only equine embryos from days 10, 12, and 14, and not from days 16 were used for Immunohistochemistry. On the opposite it, is not explained why bilaminar trophoblast pieces from day 12, 14 and 16 equine conceptuses and not from day 10 were selected for establishment of primary cell cultures.
In the Discusson, maybe due to the lack of explanation of figure 5 and the lack of figure 6, I did not came to the same conclusion of the authors that The inducible system is represented by cytosolic cPLA2, COX-2 and mPGES-1 localized in the yolk sac wall namely trophectoderm and yolk sac endoderm. Do the authors mean cPLA2 or cPGES?
Line 68: PGH2 stands for?
Line 70: Studies elucidated at least three distinct PGES
Line 72: remove there
Line 129: remove times. It should be 35 cycles.
Line 328: proven to be
Line 347: in the present study
Also all over the text please standardize mPGES-1.
References should be carefully double checked as the format differs between them.
Author Response
Comments and Suggestions for Authors
The draft “Expression of enzymes associated with prostaglandin synthesis 2 in equine conceptuses” by Budik et al. addresses a very interesting and important topic in equine maternal recognition. However, important data are not presented in the results, namely the description of figure 5, and figure 6 is not in the submitted draft thought described in the text.
Thank You for the positive general comment about the importance and novelty of the manuscript! I really apologize the missing Figure 6 it might took place due to a very inappropriate internet connection during submission. For improving the presentation of the results, we split figure 4 and of course added the missing figure (now 7).
Additionally it would improve data presentation if on figures 4, 5 and 6 authors could consider adding
- at the top of the “columns” which day/gene they correspond to;
- at the left side of the “lines” which day/gene they correspond to.
It would also help readers if those are consistent, meaning either always days at the top or at the side.
By reading, the abstract and the Introduction readers get confused:
What was indeed used for each experiment: equine conceptuses collected on days 10 (n=5), 12 (n=12), 14 (n=5) and 16 (n=7) from healthy mares for enzyme expression, and primary cell cultures derived from the yolk sac wall of equine embryos at days 10, 12, 14 and 16 to investigate the effect of oxytocin on trophoblast PGE2 synthesizing enzymes? Or both? Please clarify.
These numbers are the embryos used for quantitative RT-PCR analysis. The embryos used for primary cell culture were flushed separately!
Did the authors perform different RNA extraction for Reverse transcription qualitative PCR (RT-PCR) and Reverse transcription quantitative PCR (qRT-PCR)? If not lines 141-145 should be moved to 2.2 Extraction of RNA.
Yes, we did different RNA extraction for qualitative and quantitative RT-PCR. The RNA samples used for quantitative RT-PCR were first reversely transcribed into cDNA.
It is not explained why only equine embryos from days 10, 12, and 14, and not from days 16 were used for Immunohistochemistry.
Flushing of intact equine conceptuses is more difficult the larger the size. Therefore, we were not able to flush good quality conceptuses at day 16.
On the opposite it, is not explained why bilaminar trophoblast pieces from day 12, 14 and 16 equine conceptuses and not from day 10 were selected for establishment of primary cell cultures.
At day 10 the yolk sac wall is depending on size of the conceptus only partially bilaminar, other parts are only ectodermal.
In the Discusson, maybe due to the lack of explanation of figure 5 and the lack of figure 6, I did not came to the same conclusion of the authors that The inducible system is represented by cytosolic cPLA2, COX-2 and mPGES-1 localized in the yolk sac wall namely trophectoderm and yolk sac endoderm. Do the authors mean cPLA2 or cPGES?
Yes, the inducible system of PGE2 production consists of cPLA2-COX2-mPGES, the basal non inducible system consists of cPLA2-COX1-cPGES.
Line 68: PGH2 stands for?
Prostaglandin H2
Line 70: Studies elucidated at least three distinct PGES Done!
Line 72: remove there Done!
Line 129: remove times. It should be 35 cycles. Done!
Line 328: proven to be
Line 347: in the present study
Also all over the text please standardize mPGES-1.
We standardized!
References should be carefully double checked as the format differs between them.
Thank You we checked again!
Thank You for reviewing, happy eastern!
Reviewer 3 Report
Manuscript animals-1152609
Expression of enzymes associated with prostaglandin synthesis in equine conceptuses
The manuscript deals with an interesting and very important topic related to
the expression and localization of key enzymes of the different pathways involved in prostaglandin E2 and F2α, in the equine conceptus during the mobility phase. Results suggest that equine conceptuses possess two independent PGE2 producing systems: an inducible one (i) PLA2/COX-2/mPGES-1 localized in the yolk sac wall namely trophectoderm and yolk sac endoderm; and (ii)-PLA2/COX-1/cPGES as a constant basal one. PF2α synthesis was restricted to the “peri-embryonic” pole area and produced from PGE2 via CBR-1, due to the expression of all enzymes involved in this pathway (COX-2, mPGES-1, and CBR-1). The authors propose a model explaining how PGE and PGF2α production localized at two opposite poles of the equine conceptus facilitates equine conceptus mobility.
General comments:
This manuscript falls within the scope of the journal. The current topic is a topic of great relevance and interest to the readers of the journal. This manuscript presents as a novelty a detailed analysis of the synthesis of prostaglandins in equine embryos, which are thought to be responsible for the myometrium contractions required for embryo motility, at the time of maternal recognition of pregnancy in the mare.
The manuscript is well written, and the trial is well designed. In the Material and Methods section, some doubts are raised in the specific comments. Appropriate statistical analysis is used for mRNA data. Although results are well discussed and compared with previous publications, in the results section, data is missing, which makes it impossible to confirm the results and support the authors' conclusions.
PF2α restricted to the “peri-embryonic” pole area and produced from PGE2 via CBR-1 was not possible to confirm since IHC CBR-1 localization images are missing. In addition, positive and negative control images of IHC are also not included which would help to confirm the results obtained by the authors.
It was considered that there might be an ethical concern in this study because it is unclear if the equine embryos were collected for this study or whether embryos collected in previous studies were used. Depending on the answer, a statement that investigations have been approved by the local ethical committee may be required.
Throughout the manuscript: (i) gene names should be italicized (e.g. COX), including tables and figures; (ii) gene expression or mRNA expression should not be used to express qPCR data, as it does not measure transcription.
Specific comments
Lines 18-19- It was not possible to confirm if “Prostaglandin F2 alpha synthesis is restricted to the “peri-embryonic” pole area and relies on enzymatic conversion of prostaglandin F2 alpha” since figure 6 images are missing.
Line 26: Gene expression or mRNA expression should not be used to express qPCR data, as it does not measure transcription. Please use: “Transcripts levels, mRNA levels or RNA abundance or RNA amounts”, here and throughout the manuscript, including tables and figures.
Line 29: Here and throughout the manuscript, the gene names used should be italicized: e.g. COX.
Line 34-35: It was not possible to confirm as the figure 6 image is missing.
Line 43-44: Please rewrite since embryo motility is not unique to the equine species. What is unique is the motility during a longer period, as mention in Stout and Allen (2001). Active migration of the spherical blastocyst also occurs in ruminant and pig embryos after hatching (Bazer et al., 2009; DOI: 10.1530/REP-09-0158).
Lines 76-79- needs citation.
Lines 80-82-Please move this sentence (The effect of oxytocin on trophoblast PGE2 synthesizing enzymes was investigated in primary cell cultures derived from the yolk sac wall of equine embryos at days 10, 12 14, and 16) to after the definition of the main objective (line 88), since it was also an objective of the study.
Line 82-84: Delete. This corresponds to the results of the effect of oxytocin on trophoblast PGE2 synthesis and should not be placed here in the introduction unless all the other results are placed as well.
Line 85- The localization of enzymes involved in PGE2 was not performed in a 16-day embryo (or not shown), and those involved in PGF2 synthesis are not shown (figure 6 image is missing).
Line 88: Insert: “The effect of oxytocin on trophoblast PGE2 synthesizing enzymes was also investigated in primary cell cultures derived from the yolk sac wall of equine embryos at days 10, 12 14 and 16.”, before “Detailed knowledge of the equine…..”
Line 92- 107: Doubt: were the embryos used in this study embryos collected in 2012? If so item 2.1 was not performed in this work, and therefore the title of the item should be changed and rewritten to reflect this.
Line 104- Was any solution used to preserve the RNA of the tissues before freezing?
Line 114- Was RNA quality assessed before RT-PCR? If so, how? Why wasn’t DNAse treatment performed before sequencing?
Line 122- Gene names in table 1 should be italicized.
Line 135: How was UV detection done?
Line 152: In Table 2:
- Gene names should be italicized.
- The XM_005605567 Accession number for mPGES-1 (PTGES) in the NCBI database has been removed. Please check.
Line 167- Doubt: Was this performed in this study "Equine embryos were immersion-fixed 4% neutral buffered formaldehyde were embedded in Histo Comp" or embryos previously embedded in Histo Comp from the 2012 study were used here?
Line 190- How were samples observed and photographed? What was evaluated and how was the evaluation done?
Line 192- Doubt: the bilaminar trophoblast pieces used were from embryos obtained in this study?
Line 215-216: Citation is needed: ” ……… inactivated medium heated previously to 65 °C for 30 min in order to inactivate any endogenous oxytocin.”
Line 221- Was any solution used to preserve the RNA before freezing?
Line 228- How is qPCR data present? Mean±SD??
Lines 230- 270: Gene names used should be italicized.
Line 237- In the caption of figure 1, it says "RT-PCR of total RNA of four different equine embryos...", but in the text, it says n=1 per day, and there are no 16-day embryo data.
Line 244-247- In the qPCR results section:
-The order in which the results for the different genes are presented should correspond to the order of the graphs in figure 2.
- match the results of the different genes to the respective figures. E.g. PLA2 corresponds to fig 2.A; cPGES to fig 2B, etc.
Lines 246-247: please replace expression levels of the genes, with other words, like transcripts/mRNA abundance/levels.
Line 246: Put the PLA2 results first and before the COX results, since in figure 2 the 1st graph is for this gene: “Higher levels of PLA2 were observed on day 16than on days 10, 12 or 14 (p<0.01; see Fig. 2a).”
Lines 249-250: delete: “Higher levels of PLA2 were observed on day 16 than on days 10, 12 or 14 (p<0.01; see Fig. 2a).”
Lines 250-251: Please considered: Transcripts of cPGES and mPGES were consistently expressed in all samples, although no difference among embryo age was noted (figure 2b and 2c). These results should be presented right after the PLA2 ones.
In figure 2 a) and d) please insert the significant differences (with letters for example) in the respective plots. E.g. fig 2a), letter a should be used for day 10, 12, and 14; and letter b for day 16. In figure 2 caption (line 261) would be placed: a,b,- different letters correspond to significant differences (P< 0.01). Different letters should be used for 2d) since P is different.
In figure 2a)- what is the meaning of the letter b in 12-day data?
In figure 2d)- please delete p<0.05 among groups and put the P-value in the legend of the figure (line 261).
Lines 259-261- In figure 2:
- please consider the title: “Relative quantitative RT-PCR……”
- say how the data is presented
- put the P-value
Lines 264-266: please considered: Although transcripts of all prostaglandin E2 synthesis enzymes were detected, the presence of oxytocin in the culture medium did not alter the mRNA abundance of any enzyme examined (see figure 3).
Line 268- Please considered in Figure 3 title: “Relative quantitative RT-PCR……”
Line 270- Please considered: No significant differences among treatments were noted.
Lines 284-286- In the images of Figure 4 it is very difficult to confirm that mPGES-1 staining (K, L, M), is weaker than cPGES staining (G, H, I).
Line 286- Please correct: figure 4 K, L, M, as they corresponded to mPGES staining images. Moreover, (figure 4 K, L, M) is not well located, as these images concern mPGES. Please place in line 284, after mPGES staining.
Lines 286-295: It was not possible to confirm the results of CBR-1 since the images of figure 6 are missing.
In Figure 4 – In IHC images hematoxylin contrast is not seen. Positive and negative control images are missing and should be included.
Line 302- In Figure 5 please consider replacing “prostaglandin E2 forming enzymes”, with “COX-1 and COX-2”, since there are only images for these 2 enzymes.
In figure 5D- What is the meaning of the arrowhead?
Line 306- Images of figure 6 are missing.
Lines 333-343- Discussion and conclusions not supported by the results, as they were not presented.
Author Response
Manuscript animals-1152609
Expression of enzymes associated with prostaglandin synthesis in equine conceptuses
The manuscript deals with an interesting and very important topic related to
the expression and localization of key enzymes of the different pathways involved in prostaglandin E2 and F2α, in the equine conceptus during the mobility phase. Results suggest that equine conceptuses possess two independent PGE2 producing systems: an inducible one (i) PLA2/COX-2/mPGES-1 localized in the yolk sac wall namely trophectoderm and yolk sac endoderm; and (ii)-PLA2/COX-1/cPGES as a constant basal one. PF2α synthesis was restricted to the “peri-embryonic” pole area and produced from PGE2 via CBR-1, due to the expression of all enzymes involved in this pathway (COX-2, mPGES-1, and CBR-1). The authors propose a model explaining how PGE and PGF2α production localized at two opposite poles of the equine conceptus facilitates equine conceptus mobility.
General comments:
This manuscript falls within the scope of the journal. The current topic is a topic of great relevance and interest to the readers of the journal. This manuscript presents as a novelty a detailed analysis of the synthesis of prostaglandins in equine embryos, which are thought to be responsible for the myometrium contractions required for embryo motility, at the time of maternal recognition of pregnancy in the mare.
The manuscript is well written, and the trial is well designed. In the Material and Methods section, some doubts are raised in the specific comments. Appropriate statistical analysis is used for mRNA data. Although results are well discussed and compared with previous publications, in the results section, data is missing, which makes it impossible to confirm the results and support the authors' conclusions.
Thank You for the general positive comments about relevance and novelty! I apologize for the missing figure, that might have arrived due to the very bad internet connection I had during the first submission!
PF2α restricted to the “peri-embryonic” pole area and produced from PGE2 via CBR-1 was not possible to confirm since IHC CBR-1 localization images are missing. In addition, positive and negative control images of IHC are also not included which would help to confirm the results obtained by the authors.
We added the missing figure 6 – now figure 7 (since figure 4 was split in order to present the results better). Positive equine control images were added for COX-2, mPGES and CBR-1. Beside these Images we added also literature proofing the antibodies working in the horse. Negative IHC controls were done previously omitting the primary antibody (data not shown).
It was considered that there might be an ethical concern in this study because it is unclear if the equine embryos were collected for this study or whether embryos collected in previous studies were used. Depending on the answer, a statement that investigations have been approved by the local ethical committee may be required.
Throughout the manuscript: (i) gene names should be italicized (e.g. COX), including tables and figures; (ii) gene expression or mRNA expression should not be used to express qPCR data, as it does not measure transcription.
We used Italics for the genes and changed the labeling of the axes!
Specific comments
Lines 18-19- It was not possible to confirm if “Prostaglandin F2 alpha synthesis is restricted to the “peri-embryonic” pole area and relies on enzymatic conversion of prostaglandin F2 alpha” since figure 6 images are missing.
Figure 6 was added (now figure 7)
Line 26: Gene expression or mRNA expression should not be used to express qPCR data, as it does not measure transcription. Please use: “Transcripts levels, mRNA levels or RNA abundance or RNA amounts”, here and throughout the manuscript, including tables and figures.
We changed the labeling!
Line 29: Here and throughout the manuscript, the gene names used should be italicized: e.g. COX.
Line 34-35: It was not possible to confirm as the figure 6 image is missing.
Figure 6 (now 7) was added!
Line 43-44: Please rewrite since embryo motility is not unique to the equine species. What is unique is the motility during a longer period, as mention in Stout and Allen (2001). Active migration of the spherical blastocyst also occurs in ruminant and pig embryos after hatching (Bazer et al., 2009; DOI: 10.1530/REP-09-0158).
Was changed!
Lines 76-79- needs citation.
Lines 80-82-Please move this sentence (The effect of oxytocin on trophoblast PGE2 synthesizing enzymes was investigated in primary cell cultures derived from the yolk sac wall of equine embryos at days 10, 12 14, and 16) to after the definition of the main objective (line 88), since it was also an objective of the study.
Thank You, done!
Line 82-84: Delete. This corresponds to the results of the effect of oxytocin on trophoblast PGE2 synthesis and should not be placed here in the introduction unless all the other results are placed as well.
Thank You done!
Line 85- The localization of enzymes involved in PGE2 was not performed in a 16-day embryo (or not shown), and those involved in PGF2 synthesis are not shown (figure 6 image is missing).
Due to problems flushing day 16 equine embryos intact or at least suitable for IHC we could not perform IHC at that age, but we collected some tissue for RT-PCR. The main enzyme beside cPLA2 and COX-2 for PGF2alpha synthesis the terminal Prostaglandin F synthase (PGFS) could not be detected in any embryo by qRT-PCR. Therefore we were searching for alternative synthesis pathways and found CBR-1 expression!
Line 88: Insert: “The effect of oxytocin on trophoblast PGE2 synthesizing enzymes was also investigated in primary cell cultures derived from the yolk sac wall of equine embryos at days 10, 12 14 and 16.”, before “Detailed knowledge of the equine…..”
Thank You done!
Line 92- 107: Doubt: were the embryos used in this study embryos collected in 2012? If so item 2.1 was not performed in this work, and therefore the title of the item should be changed and rewritten to reflect this.
The embryos used for primary cell culture were flushed for that investigation, the ones flushed in 2012 were used for a study before and the samples were used for that study!
Line 104- Was any solution used to preserve the RNA of the tissues before freezing?
No they were shock frozen in liquid N2 and then stored at -80°C
Line 114- Was RNA quality assessed before RT-PCR? If so, how? Why wasn’t DNAse treatment performed before sequencing?
We assessed 260/280 nm values, DNAse treatment was performed before reverse transcription. Sequencing was done using the amplification primers using the purified DNA amplicon excised and purified from the agarose gel.
Line 122- Gene names in table 1 should be italicized.
Done
Line 135: How was UV detection done?
By Biorad Chemidoc MP using Ethidium Bromid
Line 152: In Table 2:
- Gene names should be italicized.
- The XM_005605567 Accession number for mPGES-1 (PTGES) in the NCBI database has been removed. Please check.
We changed to: NM_001081935
Line 167- Doubt: Was this performed in this study "Equine embryos were immersion-fixed 4% neutral buffered formaldehyde were embedded in Histo Comp" or embryos previously embedded in Histo Comp from the 2012 study were used here?
Line 190- How were samples observed and photographed? What was evaluated and how was the evaluation done?
Line 192- Doubt: the bilaminar trophoblast pieces used were from embryos obtained in this study?
Line 215-216: Citation is needed: ” ……… inactivated medium heated previously to 65 °C for 30 min in order to inactivate any endogenous oxytocin.”
Line 221- Was any solution used to preserve the RNA before freezing?
Line 228- How is qPCR data present? Mean±SD??
Lines 230- 270: Gene names used should be italicized.
Line 237- In the caption of figure 1, it says "RT-PCR of total RNA of four different equine embryos...", but in the text, it says n=1 per day, and there are no 16-day embryo data.
Line 244-247- In the qPCR results section:
-The order in which the results for the different genes are presented should correspond to the order of the graphs in figure 2.
- match the results of the different genes to the respective figures. E.g. PLA2 corresponds to fig 2.A; cPGES to fig 2B, etc.
Lines 246-247: please replace expression levels of the genes, with other words, like transcripts/mRNA abundance/levels.
Line 246: Put the PLA2 results first and before the COX results, since in figure 2 the 1st graph is for this gene: “Higher levels of PLA2 were observed on day 16than on days 10, 12 or 14 (p<0.01; see Fig. 2a).”
Lines 249-250: delete: “Higher levels of PLA2 were observed on day 16 than on days 10, 12 or 14 (p<0.01; see Fig. 2a).”
Lines 250-251: Please considered: Transcripts of cPGES and mPGES were consistently expressed in all samples, although no difference among embryo age was noted (figure 2b and 2c). These results should be presented right after the PLA2 ones.
In figure 2 a) and d) please insert the significant differences (with letters for example) in the respective plots. E.g. fig 2a), letter a should be used for day 10, 12, and 14; and letter b for day 16. In figure 2 caption (line 261) would be placed: a,b,- different letters correspond to significant differences (P< 0.01). Different letters should be used for 2d) since P is different.
In figure 2a)- what is the meaning of the letter b in 12-day data?
In figure 2d)- please delete p<0.05 among groups and put the P-value in the legend of the figure (line 261).
Lines 259-261- In figure 2:
- please consider the title: “Relative quantitative RT-PCR……”
- say how the data is presented
- put the P-value
Lines 264-266: please considered: Although transcripts of all prostaglandin E2 synthesis enzymes were detected, the presence of oxytocin in the culture medium did not alter the mRNA abundance of any enzyme examined (see figure 3).
Line 268- Please considered in Figure 3 title: “Relative quantitative RT-PCR……”
Line 270- Please considered: No significant differences among treatments were noted.
Lines 284-286- In the images of Figure 4 it is very difficult to confirm that mPGES-1 staining (K, L, M), is weaker than cPGES staining (G, H, I).
Line 286- Please correct: figure 4 K, L, M, as they corresponded to mPGES staining images. Moreover, (figure 4 K, L, M) is not well located, as these images concern mPGES. Please place in line 284, after mPGES staining.
Lines 286-295: It was not possible to confirm the results of CBR-1 since the images of figure 6 are missing.
No provided as Figure 7
In Figure 4 – In IHC images hematoxylin contrast is not seen. Positive and negative control images are missing and should be included. See above negative controls!
Line 302- In Figure 5 please consider replacing “prostaglandin E2 forming enzymes”, with “COX-1 and COX-2”, since there are only images for these 2 enzymes.
In figure 5D- What is the meaning of the arrowhead?
A scattered trophoblast cell within the embryonic disc!
Line 306- Images of figure 6 are missing.
Provided now as figure 7
Lines 333-343- Discussion and conclusions not supported by the results, as they were not presented.
Now obvious by presenting Figure 7
Thank You for the time consuming review, happy eastern!
Round 2
Reviewer 1 Report
Dear authors,
The manuscript presents the quality to be published.
Author Response
Thank You so much !
Reviewer 2 Report
After carefully reading the revised version of the manuscript I believe the manuscript improved considerably and is now acceptable for publication after minor corrections.
An short explanation why equine embryous from day 16 were not used for immunohistochemistry and why equine conceptuses from day 10 were not selected for the establishment of primary cell cultures should be provided.
Throughout the text genes names and abbreviations should be in italics.
Throughout the text mPGES-1 should be standardized. For example in figures 2 and 3, and lines 245 and 251.
Captation of figure 1 β-actin is not correct.
Line 275 should be figure 4 and not figure 5.
Line 278 should be figure 5 A, B, C.
Line 334 is should be "mobility proven to be important"
Line 352 shoul be "includede in the present study"
Again, references be carefully double checked:
in some references remove the symbol & (1, 21-30)
some have a dot after the last author (12, 15).
Author Response
After carefully reading the revised version of the manuscript I believe the manuscript improved considerably and is now acceptable for publication after minor corrections.
Thank You for that assessment!
A short explanation why equine embryos from day 16 were not used for immunohistochemistry and why equine conceptuses from day 10 were not selected for the establishment of primary cell cultures should be provided.
The explanations were added at lines 169/170 and 195/196
Throughout the text genes names and abbreviations should be in italics.
We checked again and hope that we changed all!
Throughout the text mPGES-1 should be standardized. For example in figures 2 and 3, and lines 245 and 251.
Captation of figure 1 β-actin is not correct.
We changed to symbol thank You!
Line 275 should be figure 4 and not figure 5.
Thank You was changed!
Line 278 should be figure 5 A, B, C.
Thank You was changed!
Line 334 is should be "mobility proven to be important"
Thank You changed!
Line 352 shoul be "includede in the present study"
Done!
Again, references be carefully double checked:
in some references remove the symbol & (1, 21-30)
some have a dot after the last author (12, 15).
We checked again and hope that it fits now!
Reviewer 3 Report
Manuscript animals-1152609- Revised Version
Expression of enzymes associated with prostaglandin synthesis in equine conceptuses
The authors answered some of the questions raised by the reviewers. The results section was improved, missing information was added, and data are properly discussed and compared with previous publications.
Some concerns should be clarified/considered by the authors:
Throughout the manuscript, there are still gene names that are not italicized.
Of the 2 citations that were requested, one was not introduced and the other is not correct:
- Lines 76-79- needs citation.
- Line 215-218: Citation is still needed for "culture medium heated previously to 65°C for 30 min inactivates endogenous oxytocin" as reference [20] provided by the authors is not a correct one. Otherwise, "in order to inactivate any endogenous oxytocin" should be deleted.
Line 102: Please delete “….and part of the conceptus material was already used in that study” since embryos used in this study were flushed for this investigation.
In table 2, NCBI database mPGES-1 (PTGES) accession number has not been changed. Please replace with NM_001081935.
Line 228- Throughout the manuscript, there is no information on how the qPCR data is presented. Mean±SD; Mean±SEM? This must be mentioned.
Line 239- How many embryos were used: three or four? In figure 1 caption is said "RT-PCR of total RNA of four? different equine embryos...", but in the text, it says n=1 per day, and no 16-day embryo data exists.
Line 247-252-The order in which the results for the different genes are presented in the manuscript should correspond to the order of the graphs in figure 2. Otherwise, change the order of the graphs in figure 2 to match the description in the text. Also, the names of the genes in the figures should be italicized.
Line 275- please replace figure 5 with figure 4, since COX1 protein in trophectoderm, endoderm corresponds to figure 4.
Line 278- COX2 corresponds to figure 5 A, B, C.
Line 303- Please mention that " D: equine endometrium COX-2"corresponds to the positive control.
Line 304: Please mention that "H: equine kidney mPGES" corresponds to the positive control.
Line 361: Please remove the parentheses from PLA2/COX-2/mPGES-1
Line 363: Please remove the parentheses from PLA2/COX-1/cPGES
Author Response
Manuscript animals-1152609- Revised Version
Expression of enzymes associated with prostaglandin synthesis in equine conceptuses
The authors answered some of the questions raised by the reviewers. The results section was improved, missing information was added, and data are properly discussed and compared with previous publications.
Some concerns should be clarified/considered by the authors:
Throughout the manuscript, there are still gene names that are not italicized.
We checked again and hope that now every gene name is italicized!
Of the 2 citations that were requested, one was not introduced and the other is not correct:
- Lines 76-79- needs citation. We add 2 new citations 13 and 14!
- Line 215-218:Citation is still needed for "culture medium heated previously to 65°C for 30 min inactivates endogenous oxytocin" as reference [20] provided by the authors is not a correct one. Otherwise, "in order to inactivate any endogenous oxytocin" should be deleted. We exchanged the citation 22!
Line 102: Please delete “….and part of the conceptus material was already used in that study” since embryos used in this study were flushed for this investigation. This part was deleted!
In table 2, NCBI database mPGES-1 (PTGES) accession number has not been changed. Please replace with NM_001081935. Was done!
Line 228- Throughout the manuscript, there is no information on how the qPCR data is presented. Mean±SD; Mean±SEM? This must be mentioned. Was changed!
Line 239- How many embryos were used: three or four? In figure 1 caption is said "RT-PCR of total RNA of four? different equine embryos...", but in the text, it says n=1 per day, and no 16-day embryo data exists. We changed the legend of Figure 1. Now it should be clear that there were 3 conceptuses 1 at day 10, 1 at day 12 and 1 at day 14!
Line 247-252-The order in which the results for the different genes are presented in the manuscript should correspond to the order of the graphs in figure 2. Otherwise, change the order of the graphs in figure 2 to match the description in the text. Also, the names of the genes in the figures should be italicized. We changed the order in the text, italicized the gene symbols and adapted the legends!
Line 275- please replace figure 5 with figure 4, since COX1 protein in trophectoderm, endoderm corresponds to figure 4.Sorry about this mistake was changed!
Line 278- COX2 corresponds to figure 5 A, B, C. Sure sorry about that mistake!
Line 303- Please mention that " D: equine endometrium COX-2"corresponds to the positive control. Was added!
Line 304: Please mention that "H: equine kidney mPGES" corresponds to the positive control.
Was added!
Line 361: Please remove the parentheses from PLA2/COX-2/mPGES-1
The parentheses were removed!
Line 363: Please remove the parentheses from PLA2/COX-1/cPGES
The parentheses were removed!
We finally thank You for the very exact review and apologize for the mistakes which happened!
Due to the ameliorations done we hope that the manuscript is now suitable for publication in “Animals”
Yours Sincerely
Sven Budik